# The UPR in Neurodegenerative Disease: Not Just an Inside Job

**DOI:** 10.3390/biom10081090

**Published:** 2020-07-22

**Authors:** Anna Maria van Ziel, Wiep Scheper

**Affiliations:** 1Department of Clinical Genetics, Amsterdam University Medical Centers location VUmc, 1081 HV Amsterdam, The Netherlands; a.m.van.ziel@vu.nl; 2Department of Functional Genomics, Center for Neurogenomics and Cognitive Research, Vrije Universiteit (VU), 1081 HV Amsterdam, The Netherlands

**Keywords:** unfolded protein response, unconventional secretion, neurodegenerative disease, proteostasis, cell non-autonomous

## Abstract

Neurons are highly specialized cells that continuously and extensively communicate with other neurons, as well as glia cells. During their long lifetime, the post-mitotic neurons encounter many stressful situations that can disrupt protein homeostasis (proteostasis). The importance of tight protein quality control is illustrated by neurodegenerative disorders where disturbed neuronal proteostasis causes neuronal dysfunction and loss. For their unique function, neurons require regulated and long-distance transport of membrane-bound cargo and organelles. This highlights the importance of protein quality control in the neuronal endomembrane system, to which the unfolded protein response (UPR) is instrumental. The UPR is a highly conserved stress response that is present in all eukaryotes. However, recent studies demonstrate the existence of cell-type-specific aspects of the UPR, as well as cell non-autonomous UPR signaling. Here we discuss these novel insights in view of the complex cellular architecture of the brain and the implications for neurodegenerative diseases.

## 1. Introduction

Neurons have an elaborate endomembrane system that underlies their unique function. In order to maintain protein homeostasis (“proteostasis”) in the neuronal endomembrane system, an intricate network of protein quality control machinery is in place [1], in which the unfolded protein response (UPR) plays a central role (reviewed in Reference [2]).

## 2. The Canonical Cell Autonomous UPR

The endoplasmic reticulum (ER) is essential for secretory and membrane proteins that are synthesized and folded at the ER before being transported to their final destination [3]. The main goal of the UPR is to restore proteostasis. If the protein folding demand exceeds the protein folding capacity in the ER, ER stress ensues and the UPR is activated. The accumulation of unfolded and misfolded proteins in the ER lumen is sensed by the ER luminal site of the three UPR signal transducer proteins, namely inositol requiring enzyme 1 (IRE1), protein kinase R (PKR)-like ER kinase (PERK) and activating transcription factor 6 (ATF6) [4] (Figure 1). Different models describe how the accumulation of unfolded proteins is sensed by the UPR transducer proteins and leads to their activation (reviewed in References [2,5]). Activation of these transducer proteins initiates downstream signaling of the three UPR pathways, resulting in an intricate transcriptional, post-transcriptional and translational network, with the overall aim to reduce the biosynthetic load (by decreasing protein synthesis and by selective mRNA degradation), increase the folding capacity (by increasing the expression of chaperone proteins) and facilitate the removal of aberrant proteins (by enhancing the proteolytic clearance capacity; reviewed in References [2,6]).

The leading theory proposes that the ER chaperone binding immunoglobulin protein (BiP, or glucose-regulated protein 78) acts as a negative regulator of UPR signaling. In the absence of ER stress, BiP binds to the luminal domains of the transducer proteins, thereby keeping them in a monomeric and inactive state. Upon accumulation of unfolded proteins, competition for BiP binding leads to dissociation from the transducer proteins. Subsequently these dimerize and oligomerize (IRE1 and PERK) [7,8,9], or translocate (ATF6) [10], thereby initiating downstream UPR signaling [5]. However, evidence from yeast indicated that BiP binding and dissociation only fine-tune UPR signaling [11,12]. Therefore, the ligand-driven model has been proposed where un- or misfolded proteins serve as activating ligands and bind the luminal domain, thereby directly activating IRE1 and PERK by stabilizing their dimerization/oligomerization [13,14,15,16]. BiP is one of the targets of which the levels increase upon UPR activation and as such also provides a negative feedback loop for the UPR. In line with this, overexpression of BiP attenuates UPR signaling [9,17]. This negative UPR regulation through BiP can be explained by both models: (1) more BiP to bind the transducer proteins, resulting in monomerization and inactivation, or (2) less unchaperoned unfolded proteins to stabilize oligomerization of the transducer proteins.

Upon oligomerization, IRE1 is autophosphorylated, thus activating both the cytosolic kinase and endoribonuclease domain. IRE1 influences protein folding load, chaperone production and protein clearance of the ER [2]. The RNase property of IRE1 is employed for the unconventional splicing of X-box binding protein 1 (XBP1) mRNA, thus removing a 26-nucleotide long intron that leads to a frameshift which creates the open reading frame of the active transcription factor XBP1s [18,19,20]. Once generated, XBP1s induce transcription of ER stress responsive genes, including the ER luminal co-chaperone and DnaJ protein ER DNAJ homolog 4 (ERdj4 or DNAJB9) [21], which is suggested to negatively regulate IRE1 signaling by recruiting BiP to luminal IRE1 [22].

The RNase activity of IRE1 also reduces protein folding load in the ER through regulated IRE1-dependent mRNA decay (RIDD) [23,24]. Upon IRE1 activation, some ER-localized mRNAs that contain a specific mRNA structure (XBP1-like stem loops; [25]) are cleaved by IRE1 and subsequently digested by 3′ to 5′ and 5′ to 3′ exoribonucleases [26]. In addition, IRE1 stimulates ER-associated degradation (ERAD) of ER-retained misfolded proteins, a process that involves translocation of misfolded proteins to the cytosol and targeting to the proteasome for degradation (reviewed in Reference [27]).

Upon misfolded protein accumulation, the stress transducer PERK oligomerizes and is auto-phosphorylated, thereby activating the cytosolic kinase domain [16,28]. Active PERK phosphorylates and inactivates eukaryotic translation initiation factor 2 α (eIF2α). This results in the transient inhibition of translation initiation of mRNAs and the reduced influx of newly synthesized proteins into the ER [29]. Paradoxically, translation is increased of selective mRNAs containing short open reading frames in their 5′ untranslated regions, including activating transcription factor 4 (ATF4 [30]). ATF4 promotes the transcription of growth arrest and DNA damage-inducible 34 (GADD34 [31,32,33]). The increased expression of GADD34 upon PERK activation creates a negative feedback loop, because it is a regulatory subunit of phosphatase 1 that facilitates the dephosphorylation of eIF2α, thus restoring translation initiation and ATF4 levels [32,33].

The third UPR transducer ATF6 is transported via coat protein complex II (COPII)-coated vesicles to the Golgi apparatus upon ER stress [34]. In the Golgi, ATF6 (90 kDa in size) is cleaved by site-1 and site-2 proteases, removing its luminal and transmembrane domain [35]. The remaining 50 kDa cytosolic ATF6 fragment is targeted to the nucleus, where it increases transcription of UPR responsive genes [17].

All three UPR branches activate transcription factors that bind to specific DNA motifs in the promoter sites of UPR responsive genes (Figure 1). XBP1s and ATF6 both regulate transcription of genes containing an ER stress element (ERSE) [20,36,37], including ER chaperones like BiP and also XBP1 itself [20,38]. However, ATF6 has a higher affinity to bind ERSE than XBP1s [37] and binding of both is dependent on the presence of nuclear transcription factor Y (NF-Y), a general non-stress-related transcription factor [20,39]. ATF6 and XBP1s together induce expression of ERAD machinery components [40,41]. Two additional UPR responsive motifs, ERSE-II [42] and the UPR element (UPRE) [43,44], are suggested to be located in promoters of ERAD machinery genes [41,44]. UPRE is bound by XBP1s homodimers and by ATF6 and XBP1s heterodimers with higher affinity [20,41]. XBP1s and ATF6 both bind ERSE-II, where only ATF6 binding is dependent on NF-Y [37]. ATF4 binds amino acid response elements (AARE1 and AARE2) and the C/EBP-ATF response element (CARE) (reviewed in Reference [45]). An important downstream target of ATF4 is the transcription factor CHOP, which in turn induces GADD34 expression.

It is important to note that the PERK, IRE1 and ATF6 branches are not separate entities, but that there is extensive crosstalk between the UPR pathways. For example, ATF4, which is regulated via the PERK pathway, increases the transcription of IRE1, thus coupling the signaling of the PERK and IRE1 pathways [46].

## 3. Cell-Type-Specific UPR Signaling

In every mammalian cell, the three UPR signaling pathways are present. However, cell-type-specific aspects of UPR signaling are observed, as well. A main driving force for the tissue-specific differences is the differential demand on ER capacity and function. For example, specialized secretory cells require a continuously high ER capacity, whereas other cell types need to quickly react to large fluctuations in protein folding load and adapt their ER capacity accordingly. In addition, specific UPR regulation is required for developmental processes in most secretory cells and affects important cellular processes, like autophagy and apoptosis [47]. Therefore, UPR signaling is fine-tuned to suit the cell-type-specific function.

Different cell types can activate different UPR branches when triggered by the same ER-stress-inducing treatment. Upon infection with the chikungunya virus, HeLa cells activate the PERK pathway; however, the same infection results in IRE1 signaling in HepG2 cells [48]. Moreover, activation of the same UPR branch can lead to a different transcriptional response in different cell types. For example, the genome-wide identification of XBP1s transcriptional targets shows differences between skeletal muscle cells and plasma or pancreatic β cells [49]. The presence of cell-type-specific UPR responses should be considered when studying the involvement of the UPR in neurodegenerative diseases that occur in the context of cellular diversity of the brain. However, differential UPR signaling in different cell types of the brain is an unexplored area of research, although differences in the magnitude and kinetics of the response have been observed when comparing primary neurons or neuroblastoma cells. For example, the XBP1s response to treatment with a pharmacological ER stressor is approximately tenfold higher in differentiated human SK-N-SH neuroblastoma cells, as compared to primary mouse neurons, whereas BiP mRNA expression levels are similar in both cell types. This is accompanied by a kinetic difference, where UPR signaling is more quickly switched off in SK-N-SH cells than in primary neurons [50]. The factors that determine the fine-tuning of the UPR response in different cells are currently unknown. Ultimately these cell-type-specific differences in UPR signaling may lead to cell-type-specific regulation of cellular responses. For instance, UPR activation affects autophagy and apoptosis differently in HeLa cells and HepG2 cells [48], but currently the potential implications of cell-type-specific UPR signaling in the brain for neurodegenerative disease are currently still elusive.

Another level of cell-type-specific UPR signaling is provided by the cell-selective expression of different extended family members of the three canonical ER stress transducers. Two mammalian IRE1 paralogs are expressed, IRE1α [51] and IRE1β [52]. Whereas IRE1α is present in various tissues, IRE1β expression is limited to epithelium cells of the intestines and airways [53,54]. With respect to neurodegenerative diseases, currently the most relevant diversity has been reported for the ATF6 family. Apart from the ubiquitously expressed two ATF6 paralogs, ATF6α and ATF6β [55], other ER stress transducer proteins with high sequence similarity to ATF6 exist. This family includes OASIS (old astrocyte specifically induced substance), BBF2H7 (BBF2 human homolog on chromosome 7) and CREB (cyclic AMP-response element-binding proteins) proteins CREBH, CREB4 and CREB3 or Luman that, just like ATF6, contain a transmembrane, basic leucine zipper (bZIP) and DNA-binding domain and can be proteolytically processed into active transcription factors (reviewed in Reference [56]). Despite these structural and functional similarities, the distribution pattern for most of these family members is more restricted to specific cell types than ATF6 [56]. OASIS and Luman are of particular interest, since they are enriched in the brain. OASIS is highly expressed in astrocytes of the central nervous system and is in fact essential for astrocytic differentiation [57,58,59]. Luman/CREB3 is highly expressed in the liver, but also in neurons in dorsal root ganglia, hippocampal CA1 and CA2, and the dentate gyrus [60]. Luman is inhibited by interaction with the UPR-regulated Luman recruitment factor (LRF), providing a selective negative feedback loop [61].

Although the role of cell-type-specific UPR signaling in neurodegenerative disease is elusive, some potentially interesting observations have been made. The extreme polarization of neurons, where the axons mediate antero- and retrograde long-distance transport may indicate the need for specialized stress responses. Neurodegenerative damage to synapses or axons may be sensed locally, but transcription factors have to be transported retrogradely to the nucleus to exert their function. ATF4 is specifically synthesized locally in axons upon treatment with β-amyloid, the major component of the extracellular protein deposits found in Alzheimer’s disease (AD) [62]. The regulation of ATF4 does not necessarily involve the UPR, because ATF4 is also upregulated by other stress pathways. Interestingly, the non-canonical UPR transducer Luman appears specifically involved in an adaptive response to axonal injury, suggesting it is a specialized neuronal ER stress transducer. Luman is upregulated in response to axotomy, and Luman deficiency impairs the regenerative capacity of the axon [63,64]. These studies indicate that, in addition to cell-type-specific responses, subcellular specialization also contributes to the fine-tuning of UPR responses.

In addition, cell-type-specific UPR signaling may affect neurodegeneration in a cell non-autonomous manner, via the activation state of microglia or astrocytes. The expression of the astrocyte enriched stress transducer OASIS is increased in reactive astrocytes, indicating that it responds to the activation state of astrocytes. Kainic acid treatment induces ER stress markers in both astrocytes and neurons. Interestingly, OASIS-deficient astrocytes are more sensitive to ER stress, whereas removal of OASIS does not affect the ER stress sensitivity of cultured neurons. However, neurons are more sensitive for kainic acid-induced toxicity in OASIS-deficient mice. Therefore, OASIS expression in astrocytes is suggested to have a cell-autonomous protective role in astrocytes, which in turn also cell non-autonomously protects neurons against kainic acid treatment [65].

In conclusion, the differential expression of ER-stress-transducer proteins, as well as differential responses of the UPR branches and target genes, contributes to cell-type-specific aspects of UPR signaling. Together, these allow customization to suit the needs of the distinct cell types regarding ER function and additional cellular processes. For neurodegenerative diseases, subcellular UPR responses and cell non-autonomous involvement of glia-specific UPR signaling potentially play a role (Figure 2). Therefore, the role of UPR signaling in disease should be studied in the disease-relevant cell type.

## 4. Cell Non-Autonomous UPR Signaling

Tissues and organs are composed of many cells and different cell types that do not function as isolated entities, but actively collaborate and communicate with each other. Recent studies have explored effects of the control of proteostasis beyond cellular borders. Cell non-autonomous proteostatic signaling was first observed for the heat-shock response (HSR). Whereas the UPR is the protein stress response pathway of the ER, the HSR performs this role in the cytosol and induces the expression of an extensive set of chaperones termed heat-shock proteins (HSP; reviewed in Reference [66]). Tissue-specific overexpression of HSP90 inhibits HSR activation in distal tissues, whereas depletion of HSP90 in specific tissues results in cell non-autonomous activation of the HSR, indicating a compensatory mechanism [67]. The cell-to-cell communication may be directly carried out by HSPs, because several of these chaperones, including HSP40, HSP70 and HSP90, can be secreted under non-stress conditions via exosomes, which can be internalized by recipient cells, thereby improving proteostasis [68]. The intercellular transmission of exosome-containing chaperones is also triggered by proteotoxic heat stress and compensates for the severely diminished proteostatic capacity in cells without a functional HSR [68].

Similar to these cell non-autonomous effects of the HSR, anticipatory cell-to-cell signaling in response to proteotoxic stress has also been proposed for the UPR [69]. Different studies suggested that the activation of all three branches of the UPR is transmissible from tumor cells (donors) in which ER stress was induced by using compounds to unstressed macrophages (acceptors) via conditioned media (CM) [70]. These results have been replicated in an additional tumor cell line combined with macrophages [71] and also in other cell types, including hepatocytic donor cells and macrophages [72] and neuronal cells and glia cells [73], all using a similar protocol. However, data from our group demonstrate that this compound-based UPR transmission protocol [70,71,72,73] is not suitable to study UPR transmission [50]. This was demonstrated in experiments where donor cells were omitted. Despite the absence of donor cells, there was still a significant increase in mRNA and protein expression of UPR activation markers in acceptor cells. This indicates that there is carry-over of the UPR-inducer in the compound-based protocol and that UPR transmission has a very minor (if any) contribution to UPR activation in acceptor cells. The observations may be explained by the high hydrophobic and lipophilic nature of commonly used compounds to induce the UPR, like thapsigargin, causing non-specific binding to, for example, cellular membranes and plastics [74,75]. Importantly, using compound-free transmission paradigms, either by nutrient deprivation or genetic tools to activate the UPR, no UPR activation in acceptor cells was observed [50]. Therefore, we find no evidence for UPR transmission defined by transmitting full UPR signaling in cell culture.

However, it is important to note that compounds like thapsigargin and tunicamycin give a robust UPR induction and are very useful to study the cell autonomous UPR. In addition, they are helpful tools to identify UPR-induced factors in culture media, facilitating research into cell non-autonomous UPR signaling (see also below), because the above-named limitations of the currently used methodology do not imply that cell non-autonomous UPR signaling does not exist.

In particular signaling through the IRE1 pathway of the UPR has been implicated in cell non-autonomous effects of the UPR. In vivo studies demonstrate that neuron-specific constitutive expression of the transcription factor XBP1s results in elevated expression of its target BiP in the intestine of *C. elegans* [76]. Similarly, XBP1s expression in pro-opiomelanocortin neurons leads to increased expression of XBP1s and target genes, including BiP, in the liver of mice [77]. The intact anatomical connections within the in vivo models between neurons and specific peripheral organs, like the intestine and liver, may be a precondition for UPR transmission. In support of the role of direct cellular contact, a study in *Drosophila* showed that cell non-autonomous UPR regulation is limited to neighboring intestinal cells. In this study, the involvement of neuronal innervation was not investigated. Knockdown of XBP1 in intestinal epithelial cells results in eIF2α phosphorylation (downstream target of a different UPR branch: PERK) in neighboring intestinal stem cells. This is not observed when XBP1 is depleted in the fat body or muscle, suggesting cell-type-specific interactions [78]. Although it should be carefully controlled whether the modulation of XBP1(s) expression is indeed restricted to the specific donor cell type, genetic intervention is less prone to artefacts than treatment with compounds. Possibly, the UPR-induced signals and the sensitivity or response to these signals may differ between tissues, as cell-type-specific differences in UPR regulation exist (as discussed above). Whether other branches of the UPR than IRE1 can modulate the UPR cell non-autonomously is still elusive.

## 5. UPR-Induced Secretion: Cargo and Potential Mechanisms

ER stress has been reported to release different proteins in the extracellular space. An example is soluble Epoxide Hydrolase (sEH), which is increased in AD brain and has a neuroinflammatory effect [79]. Pharmacological inhibition of sEH ameliorates inflammation and neurodegeneration in a mouse model for AD, indicating the relevance of secreted factors in the disease process. The control of proteostasis via UPR-induced secretion is still a relatively unexplored area of research, but some aspects of UPR-induced secretion (potentially) involved in proteostatic regulation have been identified.

Different ER chaperones have been found to be located on the extracellular side of the cell and have been implicated in cell non-autonomous functions. This includes BiP, the major HSP70 chaperone of the ER, which can translocate across the plasma membrane [80]. It was suggested to function extracellularly as a receptor that can modulate a variety of cellular processes and affect cell viability and apoptosis [81]. More recently, the co-chaperone of BiP, ErdJ3, was also shown to be secreted [82]. In addition, different members of the protein disulfide isomerase (PDI) family of ER resident thioredoxins can be identified extracellularly in an active state [83]. Extracellular PDI has been implicated in thrombus formation upon vascular injury [84]. The UPR-responsive chaperone calreticulin is also commonly found on the extracellular side of the plasma membrane of cancer cells, as well as dying cells. Extracellular calreticulin was shown to modulate the immune response [85,86,87]. Because these chaperones are all established targets of the UPR, these proteins are putative UPR-induced secretory cargo. Recently, it was demonstrated that treatment with a compound to induce the UPR changes the secretome of cultured astrocytes in a PERK-dependent manner [88]. This involves both proteins that are established secretory proteins, as well as proteins that are not normally secreted, including ER chaperones.

In the conventional secretory pathway, proteins first enter the ER, where they are then folded. When approved by protein quality control, they are transported via the Golgi apparatus before reaching their final destination: a cell organelle, plasma membrane or extracellular space [3]. However, the secretion of ER chaperones would require a specific mechanism to exit the cell, because they are actively retained in the ER. Interestingly, when the conventional ER-to-Golgi trafficking pathway is inhibited, some proteins are still secreted [89,90,91,92,93,94]. This has stimulated research into alternative secretory routes that all circumvent the Golgi, now collectively called unconventional secretion pathways [95,96,97]. Unconventional secretion is typically triggered by cellular stress, including ER stress (reviewed in Reference [98]). In turn, blocking of ER-to-Golgi transport results in ER stress and UPR activation [94,99], suggesting a role for UPR activation in the activation of unconventional secretory pathways during ER stress. It has been shown that BiP and PDI can employ both ER-Golgi-dependent and -independent secretory routes to exit the cell [100,101,102].

The unconventional secretion of many different proteins from yeast to mammals involves the Golgi reassembly and stacking proteins (GRASPs) [90,93,94,103,104,105,106]. Cellular stress, including ER stress, can trigger GRASP-dependent alternative secretion [94,98,107]. In mammalian cells, GRASP65 was first identified, later followed by GRASP55 [108,109]. Both mammalian homologs are localized at the cytoplasmic side of the Golgi membrane, where they are involved in the stacking of Golgi cisternae [108,109,110]. GRASPs tether membranes through their PDZ (post synaptic density protein 95, *Drosophila* disc large tumor suppressor and Zonula occludens-1 protein) domains located in the conserved GRASP domain [111,112]. The PDZ domains are suggested to contribute to GRASPs function in unconventional secretion, either by vesicular and plasma membrane tethering or cargo selection [94,107,113].

Interestingly, GRASP55 was shown to be regulated by the UPR. GRASP55 was reported to re-localize to the ER upon UPR activation in cell lines [114]. In primary neurons, however, this re-localization is not observed, but UPR activation increases the levels of GRASP55. This UPR-induced expression of GRASP55 can be inhibited by small molecule inhibitors of the PERK and IRE1 pathways [115]. Elevated GRASP55 levels have been described to induce unconventional secretion in mammalian cell lines [94] and *Drosophila* [91]; therefore, this potentially provides a direct mechanism in which the UPR can induce unconventional secretion in neurons. However, to date, the neuronal cargo for UPR-induced GRASP55-dependent secretion is elusive.

The connection between cellular stress and unconventional secretion is also indicated by the involvement of chaperone complexes in unconventional secretion pathways. Hsc70 and its co-chaperone DnaJc5 have been implicated in such a pathway, but it is as yet unknown whether they serve to signal to the unconventional secretory machinery, or whether they are mechanistically involved in secretion ([116]; see also below).

Another unconventional route for secretion is via exosomes. Exosomes are intraluminal vesicles that are released upon fusion of multivesicular bodies (MVB) with the plasma membrane and are therefore directly connected with the autolysosomal system. Exosomes have been shown to contain different cargo, both protein and RNA, and may thereby potentially affect cellular processes like inflammation [117]. Interestingly, cytosolic HSP40, 70 and 90 are released via exosomes when triggered by intracellular aggregation in cell and *Drosophila* models [68], indicating that exosomes contribute to the cell non-autonomous proteostatic response (see also below). Activation of the UPR stimulates MVB formation and is accompanied by increased exosome release [118]. This would potentially provide an additional mechanism for UPR-induced unconventional secretion; however, its relevance remains to be elucidated, as specific cargo for UPR-induced exosome release is currently not known.

## 6. The UPR in Neurodegenerative Disease

Neurodegenerative diseases are characterized by the accumulation of misfolded proteins in the brains of patients, indicating a severe disturbance of proteostasis in these diseases. In many neurodegenerative diseases, signs of UPR activation have been reported (reviewed in Reference [119]). Below we discuss three subclasses of neurodegenerative disorders for which evidence for involvement of the UPR in the disease mechanism has been obtained.

The most compelling case for the involvement of the UPR in the pathogenesis of neurodegenerative disease can currently be made for tauopathies. Tauopathies are a class of neurodegenerative diseases that includes AD and subtypes of frontotemporal dementia characterized by the aggregation of the tau protein. Increased protein levels of BiP were shown in the AD brain [120,121]. Later, this was supplemented with immunohistochemistry data showing increased phosphorylation (activation) of UPR markers, including elevated levels of phosphorylated-PERK (*p*-PERK), *p*-eIF2α and *p*-IRE1 in the hippocampus of AD patients, as compared to non-demented controls [121,122]. Subsequently, reports demonstrating UPR activation in affected brain areas of other tauopathies followed ([123,124,125]; reviewed in Reference [119]). In AD and primary tauopathies, UPR activation markers are closely linked to tau pathology, following a similar spatiotemporal pattern through the brain. Often, there is overlap on a single cell level, especially in neurons showing diffuse tau pathology without inclusion bodies, suggesting that UPR activation is an early event in tau pathogenesis [122,123,124,125]. Together, these findings in postmortem brain tissue suggest a connection between UPR activation and tau pathology.

Like most proteins that accumulate in neurodegenerative diseases, tau is not localized to the ER, and the tau aggregates accumulate in the cytoplasm. Since the UPR responds to misfolded proteins, specifically in the ER, it is unclear what the exact relation is between tau pathology and UPR activation, as it is likely to be caused indirectly. Protein aggregation may, for example, induce or enhance oxidative stress in neurons, which in turn may affect the UPR [126,127,128]. Some studies suggest that the accumulation of tau triggers UPR activation, including a study that provides evidence that this occurs by impairment of ERAD in cellular and mouse models [129]. In addition, UPR activation markers are observed in a subset of cells that have developed tau inclusions in mouse models of tau pathology [130,131]. In contrast to this single cell analysis, global UPR activation is not observed in a different tau mouse model, indicating that not all cells demonstrate an active UPR [132]. In this respect, it is interesting to note that the active UPR markers *p*-PERK, *p*-IRE1 and *p*-eIF2α accumulate in granulovacuolar degeneration bodies (GVBs) [133]. Recently, our group demonstrated that these are lysosomal structures that are selectively induced in neurons by intracellular tau aggregation [134], but their connection to the neuronal UPR activation status remains to be elucidated. In neuronal cells, UPR activation induces phosphorylation of tau at epitopes associated with AD [135] via PERK signaling [131,136]. A study in *C. elegans* proposes that UPR activation protects against pathological tau accumulation [137]. Further, variants of the *EIF2AK3* gene that encodes PERK [138,139,140] and polymorphisms in the *XBP1* gene [141] associated with altered transcriptional activity [142] have been identified as genetic risk factors for tauopathies. These studies support the idea that the UPR contributes to tau pathology. In summary, there are arguments in favor of UPR activation, both as cause of tau pathology and as consequence. These are not mutually exclusive, and a vicious cycle has been proposed where UPR activation enhances tau pathology and vice versa [119,131].

A second important class of neurodegenerative diseases where the UPR has been implicated involves synucleopathies that are characterized by the intracellular aggregation of α-synuclein and includes Parkinson’s disease (PD), dementia with Lewy bodies (DLB) and multiple system atrophy (MSA). Whereas in PD the aggregates accumulate in neurons, MSA mainly displays pathology in glial cells. There is only limited evidence for UPR activation in synucleopathies: *p*-PERK and *p*-eIF2α accumulate in the substantia nigra of PD [143] and MSA [144] in cells with glial cytoplasmic α-synuclein inclusions, and UPR targets are increased in LBD [145]. In experimental models using overexpression of wild-type or mutant α-synuclein, evidence for sensitization for ER stress and activation of the UPR demonstrated by increased expression of UPR targets like BiP and PDI [146,147]. Interestingly, also in human iPSC-derived neurons expressing the A53T α-synuclein mutation [148] or carrying a α-synuclein triplication [149], the UPR is induced. Both genetic models result in familial early onset PD and are therefore an indication that the UPR may be involved in the disease mechanism of PD. Because α-synuclein like tau forms aggregates in the cytosol, its effect on the UPR is also indirect. Moreover, α-synuclein aggregates were reported to induced the UPR by interfering with ER-Golgi transport of proteins [146]. In the scope of this review, this is an interesting observation, because, as described above, the blocking of conventional secretion is a trigger for unconventional secretion regulated by the UPR (see further below).

Prion diseases or transmissible spongiform encephalopathies, including Creutzfeldt Jakob Disease (CJD), are fatal neurodegenerative disorders characterized by extracellular accumulation of the scrapie form of the prion protein (PrP^Sc^), a pathological isoform of the normal cellular prion protein (PrP) resulting in rapidly progressing neuronal loss. Although UPR markers GRP58, BiP and GRP94 were shown to be increased in CJD [150], *p*-PERK, *p*-eIF2α and *p*-IRE1 were only demonstrated in case of tau co-morbidity [125,151]. Prion disease in mouse appears to selectively activate the PERK pathway of the UPR [152]. Interestingly, recent evidence indicates that the involvement of the UPR in mouse prion disease may be mediated via astrocytes and not neurons [88]. Glial cells mediate neuroinflammation, which is under direct control of the UPR [153,154]. Although the extracellular location of the prion aggregates may therefore involve a different UPR-mediated disease mechanism than the intracellular aggregates found in the majority of neurodegenerative proteinopathies, the inflammatory response is an important pathogenic mechanism in other neurodegenerative diseases, as well. It is therefore important to define whether UPR activation in neurodegenerative diseases occurs in neurons or in glia to distinguish cell autonomous and cell non-autonomous effects on neuronal degeneration.

Over the last decade, an important cell non-autonomous mechanism shared by neurodegenerative proteinopathies has been identified. Prion disease is characterized by the propagation of the misfolded state of PrP^Sc^ to PrP, which underlies the transmissible properties of the prion pathology. More recently, it was shown that pathological assemblies of other disease-causing proteins, including tau and α-synuclein, can propagate in a similar manner. This is thought to be mediated by a prion-like seeding mechanism, where pathological tau or α-synuclein fragments (“seeds”) enter neurons and induce misfolding of endogenous tau and α-synuclein, respectively. This results in increased aggregation in the recipient neuron and in turn generates additional seeds capable of triggering misfolding [155,156]. For tau pathology, this mechanism was suggested to spread pathology from cell-to-cell via synaptically connected brain regions and underlie the progression of neurodegeneration and symptoms in AD [156,157]. The UPR has been investigated as a therapeutic target in different neurodegenerative diseases (reviewed in References [158,159,160]). Both stimulation and attenuation of the PERK and IRE1 signaling branches of the UPR are reported to have beneficial effects in mouse models of neurodegenerative diseases (reviewed in References [119,158,160]). This could be due to differences in UPR activation states between different models. Acute UPR activation is an adaptive and protective response; however, prolonged activation is maladaptive and has detrimental effects [119]. Therefore, detailed understanding of the UPR activation status and its role in the disease pathogenesis is essential for proper therapeutic intervention.

## 7. Cell Non-Autonomous UPR Signaling in Neurodegenerative Disease

The proteostatic defect, and thus the accumulation of misfolded proteins in most neurodegenerative diseases, primarily arises in neurons, but some diseases (e.g., MSA) show accumulation preferentially in glia cells [161,162]. Moreover, if the pathology is exclusively present in neurons, glia cells are typically actively involved in the pathological phenotype, for example by eliciting an inflammatory response. In both cases, cell non-autonomous mechanisms play an important role (e.g., [163]). Moreover, the concept that the pathology spreads throughout the brain via a cell non-autonomous prion-like propagation mechanism is currently a leading hypothesis [164]. Because the UPR is investigated as therapeutic target for neurodegenerative disease, the ramifications of UPR activation, including UPR-induced secretion and its cell non-autonomous effects, should be considered. Below, the putative function of proteostatic-stress-induced secretion in intra-, extra- and transcellular proteostasis is discussed (see also Figure 3).

### 7.1. Intracellular Proteostasis

Secretion induced by proteotoxic stress could function to prevent protein overload in ER and cytosol, thereby ensuring intracellular proteostasis. This may be caused by, for example, insufficient capacity or impairment of protein quality control, which occurs during intracellular protein aggregation or aging. In line with this hypothesis, cells adapt to the loss of proteasomal degradation by selectively secreting misfolded proteins, using the ER-stress-induced and ER-anchored [165] ubiquitin-specific protease 19 (USP19) for cargo selection [166]. Secretion of the PD-associated α-synuclein is also observed via this USP19-dependent route [166]. In addition, different proteins involved in neurodegenerative proteinopathies, including α-synuclein, tau and TDP43, are secreted via an unconventional pathway that employs Hsc70 and its co-chaperone DnaJc5 [116].

When the conventional ER-to-Golgi secretory pathway is blocked, cellular stress pathways are activated, thus triggering most of the identified unconventional pathways (reviewed in [98]). This is illustrated by a study showing the increased trafficking of plasma membrane proteins via a non-conventional route when the canonical secretory pathway is impaired [89,94]. Moreover, unconventional secretion of lysozyme is triggered when the classical pathway is impaired by disruption of the ER-Golgi structure during bacterial infection [167]. In all three studies, the non-conventional secretion triggered by the disruption of the conventional secretory pathway is probably mediated through ER stress and the UPR, since alternative ER-stress-inducing treatments also induce unconventional secretion [89,94,167]. These results suggest that the UPR is involved in triggering unconventional secretion to bypass a block in the canonical secretory route, thereby preventing protein accumulation in the ER. This scenario infers a specific vulnerability of neurons, where axonal transport puts great demands on the conventional secretory pathway.

Collectively, this suggests that, upon disturbances in proteostasis, adaptive secretory routes are activated to remove accumulating or misfolded proteins. Therefore, this could be an alternative pathway to maintain or restore intracellular proteostasis when classical protein quality control systems are insufficient [166] or conventional secretion is blocked. Importantly, this protective mechanism may potentially turn toxic as it contributes to the spreading of aggregation-prone proteins in neurodegenerative diseases via cell-to-cell transmission of pathological “seeds” [166,168,169]. Unconventional secretion is involved in the secretion of aggregation-prone proteins, as was demonstrated for α-synuclein and β-amyloid [116,169,170,171], but the specific mechanism requires further study. Interestingly, exosomes have also been shown to contain pathological proteins involved in neurodegenerative disease and may therefore contribute to the spreading of neurodegenerative pathology (e.g., [172,173,174]). Moreover, inhibition of exosome release inhibits propagation of pathology in a tauopathy model, suggested to be mediated by microglia-derived exosomes [175]. Currently it is elusive whether the low levels of exosomes that are released and the diverse variety of cargo that they contain allow direct effects in the spreading of pathological seeds or whether they play a more indirect regulatory role in disease pathogenesis.

### 7.2. Extracellular Proteostasis

As was first hypothesized for the HSR, the UPR has also been proposed to prevent or resolve extracellular proteostatic disturbances by inducing secretion of chaperones. UPR activation, more specifically expression of active ATF6, induces the secretion of the HSP40 BiP co-chaperone ERdj3 either alone or co-secreted bound to a misfolded client protein. Extracellular ERdj3 binds misfolded proteins and inhibits their aggregation, thereby facilitating extracellular proteostasis [82]. In contrast, UPR activation can also reduce secretion of misfolded proteins and, in this way, attenuate extracellular aggregation, as is observed for the amyloidogenic immunoglobulin light chain upon stress-independent ATF6 and XBP1s activation [176].

### 7.3. Transcellular Proteostasis

ER stress can also induce a particular type of intercellular communication: danger signaling. This involves the secretion of damage-associated molecular patterns (DAMPs) to alert recipient cells or tissues of upcoming cellular stress, for example, to activate an immune response (reviewed in [177]). In macrophages, ER stress is shown to induce the secretion of pro-inflammatory cytokines via ATF4 [178] or XBP1s signaling [179]. Interestingly, the secretion of pro-inflammatory cytokine interleukin-1β (IL1β) is mediated through GRASP55 [90]. Sustained IRE1 activation leads to secretion of cytokines via its RIDD activity. It stabilizes thioredoxin-interacting protein (TXNIP) via degradation of a regulatory miRNA, in turn resulting in activation of the NOD-like receptor pyrin domain containing-3 (NLRP3) inflammasome and IL1β secretion [180]. In addition, the accumulation of RNA products from the RIDD activity of IRE1 induces the secretion of interferon [181]. Although also mainly studied in the context of immunomodulation in the periphery, recently it was shown that extracellular calreticulin facilitates the phagocytosis of pathogens by the immune cells of the brain, the microglia [182]. It is tempting to speculate that calreticulin is a UPR signal derived from proteostatically challenged neurons to stimulate microglial clearance cell non-autonomously, but this awaits further validation.

In addition, the concept of danger signaling may be extended to other stress-related signals that enable recipient cells to adapt to upcoming changes in their proteostatic environment. Not only can protein stress induce secretion to regulate extracellular proteostasis, but it may extend its reach to neighboring cells and even other tissues. This could lead to a more coordinated response and possibly increase resistance against protein stress on an organismal level (reviewed in References [66,183,184]) and has been named the anticipatory UPR [69]. This has been demonstrated for cytosolic chaperones that are secreted upon proteotoxic stress and improve proteostasis in recipient cells and tissues ([67,68,185]; also see above). Similar mechanisms may apply to regulation of transcellular proteostasis via the UPR, where genetically increased XBP1 expression was shown to affect transcellular UPR activation and resistance to ER stress in *C. elegans*, *Drosophila* and mice [76,77,78].

The secreted signals do not necessarily have a positive effect, but may contribute to the neurodegenerative phenotype. For example, secreted astrocyte-derived factors were suggested to underlie the cell non-autonomous involvement of the UPR in neurodegeneration in mice with prion disease [88]. Whether this is caused by the observed changed astrocytic secretome awaits further study, as the effect was addressed by conditioned medium transfer, using thapsigargin to induce the UPR that is prone to artefactual carry-over, as discussed above [50].

Although stated as separate options, the hypotheses above are not mutually exclusive. For example, the secretion of waste like aggregation-prone proteins could be accompanied by secretion of stress-induced danger signals or chaperones. ER stress and UPR activation may be the common trigger for a non-conventional secretory pathway and stimulate the secretion of a wide range of proteins serving multiple proteostatic functions. However, it is possible that the degree of protein stress and other cellular disturbances fine-tune the secretory profile through differential UPR activation. In view of pathology, the removal of the aggregation-prone proteins may be protective at a cell autonomous level; however, it could contribute to disease progression via transcellular spreading of pathology. Secretion of chaperones, as well as an anticipatory UPR, offers potential mechanisms by which ER stress can counteract extracellular proteotoxicity and limit the spreading of neuropathology.

## 8. Concluding Remarks

Recently, it has become clear that there is more to UPR signaling than the cell autonomous signaling of the three canonical UPR pathways. Cell-type-specific UPR signaling and the presence of specific subcellular responses that have been observed in neurons add multiple levels of complexity, underscoring the importance to study signaling in the disease-relevant cell type. In addition, accumulating evidence indicates that the UPR has cell non-autonomous effects. These may have a major impact on neurodegenerative disease pathogenesis via UPR-driven glial inflammatory responses, but also by, for example, affecting the propagation of protein pathology from neuron-to-neuron. To obtain more insight into the cell non-autonomous signaling of the UPR in neurons, characterization of the UPR-induced secretome is essential to identify neuronal cargo. An additional interesting possibility is the use of UPR-secreted proteins as a biomarker for diagnosis, prognosis and treatment-monitoring of neurodegenerative diseases.

Preclinical studies have indicated the potential of targeting of the UPR for treatment of neurodegenerative diseases [158,159,160]. Although these hold a promise for the future, caution is warranted regarding clinical studies. For example, adaptive UPR-induced secretion of toxic protein assemblies may at some point contribute to the spreading of disease and worsen rather than ameliorate the disease. Given the current knowledge, the net outcome of interventions in the intricate cell autonomous and cell non-autonomous UPR signaling network is difficult to predict. Targeting UPR-mediated processes for treatment of neurodegenerative proteinopathies is an attractive strategy, as it employs the cell’s own machinery to fight proteostatic problems. To date, however, the UPR has outwitted the simplified schematics of scientists. This indicates a need for more fundamental knowledge of one of the most fascinating adaptive cellular pathways before the step to the clinic can be made.

## Figures and Tables

**Figure 1 biomolecules-10-01090-f001:**
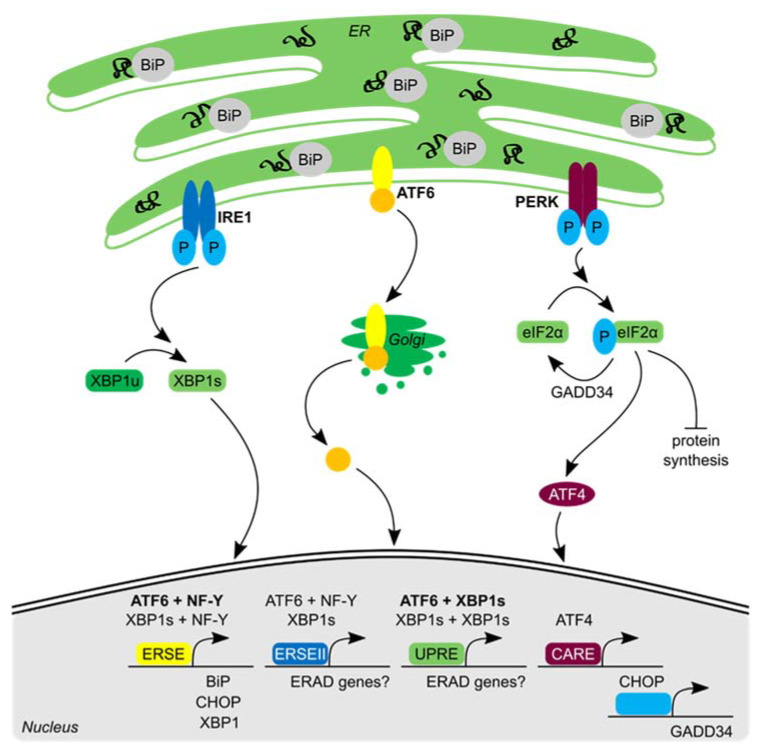
Signaling and gene regulation of the canonical cell autonomous unfolded protein response (UPR). The accumulation of unfolded proteins in the endoplasmic reticulum (ER) induces the three signaling cascades initiated by the transmembrane proteins: IRE1, ATF6 and PERK. This results in the inhibition of protein translation through PERK and the production of active transcription factors: XBP1s, ATF6 and ATF4 via IRE1, ATF6 and PERK, respectively. Upon translocation to the nucleus, XBP1s and ATF6 bind in different compositions to specific promoter sequences, including ERSE, ERSE-II and UPRE, and initiate transcription of multiple UPR genes, including ER chaperones and ERAD genes. Nuclear ATF4 binds the CARE promoter motif and induces CHOP transcription, which in turn increases GADD34 levels, thereby providing negative feedback in the PERK pathway. For details see main text.

**Figure 2 biomolecules-10-01090-f002:**
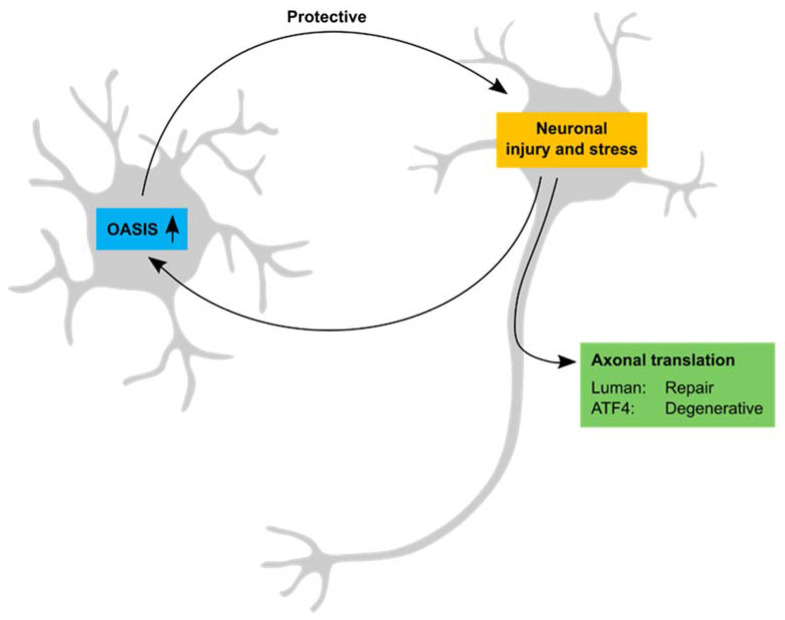
Potential interaction of cell-type-specific UPR responses in neuronal stress and injury. Neuronal stress (i.e., kainate or Aβ toxicity) or injury (i.e., axotomy) induces the UPR in glia cells and neurons. In addition to the canonical UPR present in every cell, cell-type-specific and cell non-autonomous signaling are activated. The astrocyte-specific UPR transducer OASIS elicits both a cell autonomous and cell non-autonomous protective response. In the neuronal axon, the transcription factor ATF4 is selectively upregulated and mediates a degenerative response, although the involvement of the UPR is not conclusively demonstrated. The neuron-specific UPR transducer Luman/CREB3 is specifically translated in the stressed axon and elicits a cell autonomous protective response. The combination of protective and degenerative cell autonomous and cell non-autonomous UPR responses will ultimately determine the fate of the neuron: degeneration or survival. Separate pathways have been shown to exist; the interactions between cells and pathways are hypothetical. See text for further detail.

**Figure 3 biomolecules-10-01090-f003:**
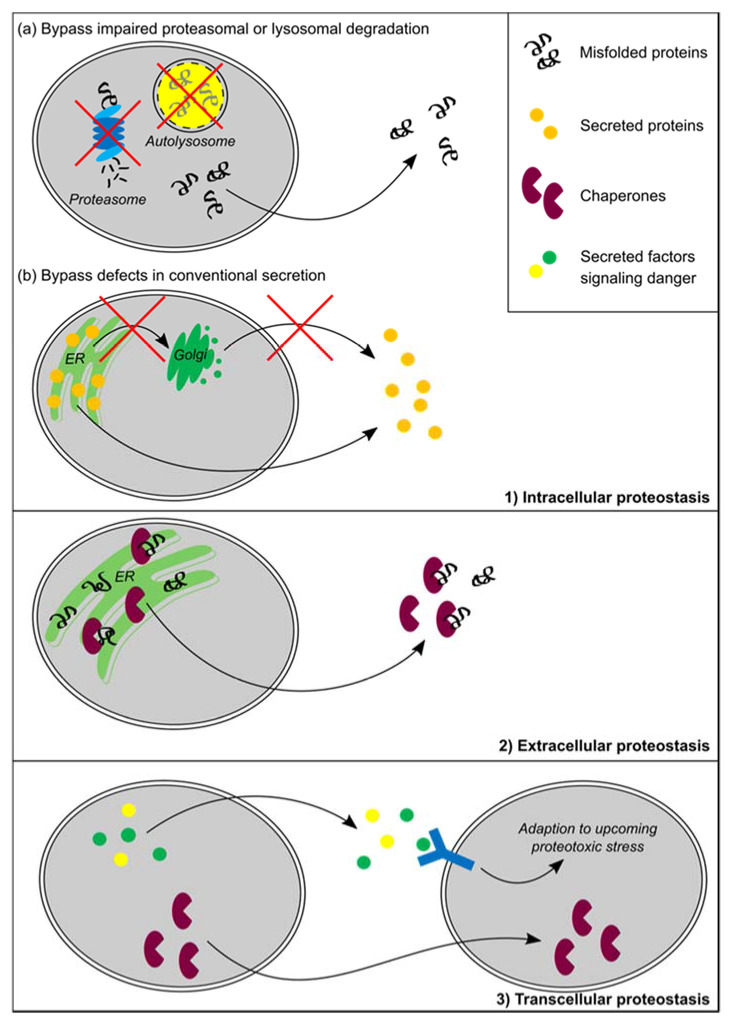
Putative proteostatic functions of cell non-autonomous UPR signaling. Schematic depiction of three hypotheses showing possible functions of cell non-autonomous UPR signaling. (**1**) Upon protein stress, accumulating or misfolded proteins from ER or cytosol are secreted to maintain intracellular proteostasis. For example, impairment of proteasomal or autolysosomal degradation (**a**) or defects in the conventional secretory pathway (**b**) have been observed to cause protein stress and induce protein secretion. (**2**) ER chaperones are secreted alone or together with misfolded proteins, to maintain or restore proteostasis in the extracellular space. (**3**) Cells that experience proteotoxic stress can secrete “danger signals” or DAMPs to alert recipient cells for upcoming proteotoxic insults. In addition, chaperones can be secreted to compensate chaperone levels between cells. Both forms of transcellular communication can contribute to proteostasis at the tissue or organismal level. For details, see main text.

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
