# Peer review of "The UPR in Neurodegenerative Disease: Not Just an Inside Job"

_biomolecules, 2020, doi:10.3390/biom10081090_

Round 1

Reviewer 1 Report

In the review by Anna Maria van Ziel and Wiep Scheper entitled “The UPR in neurodegenerative Disease: not just an inside job”, the authors describe the canonical UPR and then go on to examine UPR signaling in the context of cell type, regulation of secretion and finally maintenance of intracellular, extracellular and transcellular proteostasis.  The authors discuss these mechanisms in the context of the pathogenesis of neurodegenerative disease although this is limited to tauopathies.

Overall, the review is interesting and well written but it lacks the focus on neurodegenerative disease that the title implies.

Specific comments:

1.     The introductory paragraph is very similar to the Abstract and should be re-written or removed.

2.     The “Canonical cell autonomous UPR” section is well written and informative and furnished with a Schematic describing signaling through all three UPR branches. The schematic should include C/EPB Homologous Protein (CHOP) in the PERK pathway.

3.     The “Cell type specific UPR signaling” section lacks focus on neurodegenerative disease and on specific signaling outcomes resulting from the differential expression of genes and isoforms that are described. This section should be re-written.

4.     The manuscript contains multiple examples where protocols used by multiple investigators are criticised for using conditioned culture medium to investigate transmission of stress signals across cell types. Although the authors may have concerns about the methods used in these studies, in several of the cited examples, the authors have gone to great lengths to account for “carry-over of pharmaca” including extensive washing procedures and mass spectrometry analysis of the culture medium.  It is not reasonable or helpful to the reader to dismiss the findings of these studies in this way. These sections of the text should be re-written in a more balanced and guarded way.

5.     The concluding remarks are not strong and should be used to pull the main points of the manuscript into focus and offer some perspective on the key questions and future directions of this area of work in the context of neurodegenerative disease.

Minor:

Consider replacing the term “pharmaca”

Author Response

1. The introductory paragraph is very similar to the Abstract and should be re-written or removed.

We have replaced the introductory paragraph to avoid repetition.

2. The “Canonical cell autonomous UPR” section is well written and informative and furnished with a Schematic describing signaling through all three UPR branches. The schematic should include C/EPB Homologous Protein (CHOP) in the PERK pathway.

CHOP has been included in the revised Figure 1.

3. The “Cell type specific UPR signaling” section lacks focus on neurodegenerative disease and on specific signaling outcomes resulting from the differential expression of genes and isoforms that are described. This section should be re-written.

We agree with the reviewer that this section needed rewriting. There is limited information available with regard to neurodegenerative disease, but we have restructured and included more information to obtain a clearer overall message.

4. The manuscript contains multiple examples where protocols used by multiple investigators are criticised for using conditioned culture medium to investigate transmission of stress signals across cell types. Although the authors may have concerns about the methods used in these studies, in several of the cited examples, the authors have gone to great lengths to account for “carry-over of pharmaca” including extensive washing procedures and mass spectrometry analysis of the culture medium.  It is not reasonable or helpful to the reader to dismiss the findings of these studies in this way. These sections of the text should be re-written in a more balanced and guarded way.

We understand the reviewer’s remark, because we ourselves found these studies convincing before we started to use it in a controlled set-up. However, we do feel it is important that scientists are aware of its limitations. It is intuitively difficult to see that in the indeed extensive washing procedures would not remove the stressors, but that is the case with these compounds that are poorly soluble in aqueous solution. Also, the at glance highly convincing mass spectroscopy experiment (Mahadevan et al., 2013) demonstrating that Thapsigargin is no longer present after washing can unfortunately not be taken as evidence for that reason. We have repeated this experiment with an experienced analytical chemist. This demonstrated that a large amount of Thapsigargin is retained within the mass spec system (again related to its insolubility). The absence of Thapsigargin in culture medium can therefore not be taken as evidence that it has been removed by the washing procedure (and in fact our other biological experiments show that it has not been removed completely, “UPR transmission” can be obtained using an empty culture dish without cells; Van Ziel et al., 2019).  To better explain the reason for our concerns with these protocols we have now better specified the evidence demonstrating that this protocol is highly artefact-prone. This includes evidence that the extensive washing procedures are insufficient to remove the stressors used, the associated limitations to reliably detect such stressors in culture medium using mass spectroscopy and the absence of UPR transmission if the UPR is induced in protocols that do not employ these pharmaca, by genetic tools or nutrient deprivation. Despite these limitations of the use of conditioned media in this protocol, the approach is useful to identify secreted tUPR factors. We have made this distinction clearer in the revised manuscript.

5. The concluding remarks are not strong and should be used to pull the main points of the manuscript into focus and offer some perspective on the key questions and future directions of this area of work in the context of neurodegenerative disease.

We agree with the reviewer and have rewritten the concluding remarks.

Minor:

Consider replacing the term “pharmaca”

We have replaced “pharmaca” with “compound(s)” at the reviewer’s suggestion.

Reviewer 2 Report

This manuscript reviews the unfolded protein response (UPR) system with some emphasis on neurodegenerative diseases. The UPR is an ubiquitous system to protect cells from misfolding proteins in the endoplasmatic reticulum. First, the UPR pathway is described in detail. Then, cell type-specific UPR signaling is discussed. Next, the authors try to establish a link from UPR to neurodegenerative diseases using the example of tauopathies. However, since it is not clear whether UPR activation is the result of aberrant protein production or whether UPR activation contributes to misfolded protein pathology, this section remains rather vague and not so convincing. Next, the topic of cell non-autonomous UPR signaling is discussed and - based on the work of the authors - largely rejected as a proven mechanism. Before this topic is taken up again in relation to neurodegenerative diseases, UPR-induced secretion is discussed. May be, this paragraph might be better placed before discussing neurodegenerative diseases. This could eventually provide the opportunity to discuss the principal topic of this review (neurodegenerative diseases) in one coherent section rather then two. For my taste, the general description of the UPR could be reduced to the essential points leaving more space for the discussion of its potential role in neurodegenerative diseases (not only tauopathies).

Author Response

We thank the reviewer for these helpful comments: We have changed the order of the chapters according to the reviewer’s suggestion and extended the section on neurodegenerative disease to include discussion on synucleopathies and prion disease. This section also elaborates further on cause and consequence with regard to protein aggregation and UPR. 

Reviewer 3 Report

Dear authors,

your review entitled "The UPR in neurodegenerative Disease: Not Just an Inside Job" from Anna Maria and Wiep. I found a really interesting review and quite important at this moment for neuroscience. However, I found that there are substantial improvements that you have to do it in order to obtain a comprehensive and critical view necessary for every review needs.

Taking into account this main message or idea I can propose differents things that maybe can help you to improve the review. So, the idea is clear but as you know the UPR signalling as usually at the molecular level offers some controversy regarding the results.

Minor concerns:

-English grammar and acronym for every protein that you explain.

-You have not always explained before the situation of a protein that appears in the review and that makes it difficult to follow the thread.

-Failed some more figure some table that summarize the experiments to validate the evidence.

-Regarding the figure 1, although I can understand that is a summary try to add the proteins or genes which you are talking in the review.

-I like the section cell type-specific UPR signaling in which you said "Therefore, UPR signaling is fine-tuned to suit the cell type-specific function". After that, you go directly to the UPR in neurodegenerative disease. On the one hand, I think is important to explain what happens in the neurons and glia, normally like a little section with some figure because for me, as a neuroscientist who works in neuropharmacology it is important to know which combination of markers can explain improvement in the neuron and which ones explain neurodegeneration.

-In the next section, you only focus in tauopathies but I think you can do some efforts to include more causes.

-Add inflammation and Oxidative stress as a cause of UPR modulation. It is important because the state of the art of the AD is changing to these events.

-Reorganize the experimental studies and add tables from pharmacological studies in c elegans and rodents for example. 

-Check the literature and add more of them. For example:

Griñán-Ferré et al., 2020 "Pharmacological inhibition of soluble epoxide hydrolase in AD in which it is described the relation of inflammation and UPR signaling. You can reference in your review, just as an example of pharmacological intervention for AD in rodents that modifies UPR signaling through the modulation of inflammation.

-I think the review can be accepted after these changes.

Thank you so much.

Author Response

Minor concerns:

-English grammar and acronym for every protein that you explain.

We have checked English grammar/spelling and updated the introduction of abbreviations in the revised version

-You have not always explained before the situation of a protein that appears in the review and that makes it difficult to follow the thread.

We have optimized the introduction of abbreviations in the revised version.

-Failed some more figure some table that summarize the experiments to validate the evidence.

We have included a new figure 2 that visualizes relevant cell type specific UPR transducers and cell autonomous and cell non-autonomous effects in neurons and astrocytes. 

-Regarding the figure 1, although I can understand that is a summary try to add the proteins or genes which you are talking in the review.

We have included CHOP as major downstream UPR target and its respective downstream target GADD34 in Figure 1. This figure illustrates the canonical UPR, which we think is useful for less informed readers. We have included a new figure 2 that visualizes relevant cell type specific UPR transducers.  

-I like the section cell type-specific UPR signaling in which you said "Therefore, UPR signaling is fine-tuned to suit the cell type-specific function". After that, you go directly to the UPR in neurodegenerative disease. On the one hand, I think is important to explain what happens in the neurons and glia, normally like a little section with some figure because for me, as a neuroscientist who works in neuropharmacology it is important to know which combination of markers can explain improvement in the neuron and which ones explain neurodegeneration.

We agree with the reviewer that this section needed rewriting. There is limited information available with regard to neurodegenerative disease, but we have restructured and included more information to obtain a clearer overall message. In addition, we have included a new figure 2 that visualizes relevant cell type specific UPR transducers. 

-In the next section, you only focus in tauopathies but I think you can do some efforts to include more causes.

At the reviewer’s suggestion we have extended this section and included discussion on synucleopathies and prion disease. 

-Add inflammation and Oxidative stress as a cause of UPR modulation. It is important because the state of the art of the AD is changing to these events.

We thank the reviewer for this suggestion and have now included discussion on the role of oxidative stress and inflammation in the role of the UPR in neurodegenerative disease in the revised section on neurodegenerative disease.

-Reorganize the experimental studies and add tables from pharmacological studies in c elegans and rodents for example. 

The number of studies is too limited to warrant a table, but we have extensively rewritten and restructured the manuscript to facilitate readers.

-Check the literature and add more of them. For example:

Griñán-Ferré et al., 2020 "Pharmacological inhibition of soluble epoxide hydrolase in AD in which it is described the relation of inflammation and UPR signaling. You can reference in your review, just as an example of pharmacological intervention for AD in rodents that modifies UPR signaling through the modulation of inflammation.

We have included this reference as an excellent example of UPR induced secretion that regulates cellular processes. In addition, we have made more explicit that the focus of this review is on UPR-induced secretion that directly affects proteostasis.

Round 2

Reviewer 1 Report

The authors have addressed clearly the points which were raised in the first review. Overall the manuscript has more focus and is improved.  However, although improved, the “Cell type specific UPR signaling” section remains the weakest part of the manuscript and still lacks real focus on neurodegeneration.  The Authors do offer additional clarification of their views on the use of conditioned culture medium to investigate transmission of stress signals across cell types.  I do not doubt that their concerns may be valid or that this protocol may be prone to artefact.  However, it would be helpful to the reader if the authors could try to qualify their comments further. For example, the primary reason given for the inconsistency is the solubility of the compounds. If a compound is insoluble to the point that it cannot be removed by washing or identified by mass spectrometry it would surely have limited or no biological effect?   The authors should support their views by referring to studies which have investigated the solubility of these compounds at the relevant concentrations.

Author Response

  1. We have adapted the text of the chapter on cell type-specific UPR signaling to more clearly indicate that although this important when considering the role of the UPR in neurodegenerative diseases, this is an emerging field and that this has hardly been studied in the cells of the brain. 
  2. In the revised version we attempted to clarify our concerns with UPR transmission using conditioned media of compound treated cells. We agree with the reviewer that we have not fully succeeded in this effort. We have adapted the text thereby focusing more on the point that we would like to make: that the protocol is artefact prone and not suitable to address this and the biological evidence to support this. As likely explanation for the artefact, we have now rephrased solubility in aqueous solution to hydrophobic and lipophilic properties and provide references for the high level of non-specific interactions of the thapsigargin compound because of these properties.